# Machine learning of Ising criticality with spin-shuffling

Pallab Basu$^{a}$ ♣,$^{*}$ Jyotirmoy Bhattacharya$^{b}$ ♣,$^{†}$
Dileep Pavan Surya Jakka$^{b}$ ♣,$^{‡}$ Chuene Mosomane$^{a}$ ♣,$^{§}$ and Vishwanath Shukla$^{b}$ ♣¶

$^{a}$*Mandelstam Institute for Theoretical Physics, University of Witwatersrand, Johannesburg, South Africa.*
$^{b}$*Department of Physics, Indian Institute of Technology Kharagpur, Kharagpur 721302, India.*

We investigate the performance of neural networks in identifying critical behaviour in the 2D Ising model with next-to-nearest neighbour interactions. We train DNN and CNN based classifiers on the Ising model configurations with nearest neighbour interactions and test the ability of our models to identify the critical (cross-over) region, as well as the phases in the presence of next-to-nearest neighbour interactions. Our main objective is to investigate whether our models can learn the notion of criticality and universality, in contrast to merely identifying the phases by simply learning the value of the order parameter, magnetization in this case. We design a simple adversarial training process which forces our models to ignore learning about the order parameter while training. We find that such models are able to identify critical region with reasonable success, implying that it has learned to identify the structure of spin-spin correlations near criticality.

## I. INTRODUCTION

From the physics inspired neural nets to machine learning (ML)[1] of ground state wave functions, recently there has been a significant success of artificial neural nets (ANN) to learn the complex behaviour of physical systems [1]. In fact, over the last couple of decades, the similarity of the structure and functioning of ANNs with concepts in physics, such as RG flows [2, 3], has been repeatedly emphasized in the literature. For many of these studies, spin models in physics have served as a useful experimental playground. In recent year, both unsupervised learners like restricted Boltzmann machines [4, 5] and supervised learners like convolutional neural networks (CNNs) [6, 7], have been used to identify phases of spin models. In particular a lot of ML related experiments have been performed on the Ising model [3, 8–14]. It has been demonstrated that ANNs can successfully distinguish low and high temperature phases of the Ising model with reasonable accuracies [8]. Application of ANN in conformal field theories have been discussed in [15].

In this work, we concentrate on second order phase transition and focus on the Wilson-Fisher fixed point, realized in nearest neighbor ($\mathcal{NN}$) and also next to nearest neighbor ($\mathcal{NNN}$) Ising model. Our final goal would be to teach ANNs the concept of criticality and universality. Now since ANNs can handle only a finite vector, with at best a few thousand entries, we shall confine ourselves to a lattice of size $40 \times 40 = 1600$. However, the concept of a phase and phase transition exists only for an infinite lattice [16]. We address this issue by defining a suitable critical or cross-over region around the critical temperature $T_c$. A spin configuration at a particular temperature is labeled accordingly as critical or non-critical. We then investigate whether a trained ANN can identify this critical region. The same lattice configuration may come from two close-by but different temperatures with finite probabilities. Hence, any such phase labeling is inherently probabilistic and ML should be able to handle it.

Moreover to imbibe the the spirit of universality, we employ a different train and a test dataset corresponding to spin-models in the Ising universality class, which are closely related but still *different*. More specifically we choose the $\mathcal{NN}$ data-set to train and $\mathcal{NNN}$ data-sets to test the performance of the ANNs. Although, a lot of work has been already done on the prediction of phases in the Ising model, this investigation in the presence of $\mathcal{NNN}$ interactions has never been reported in the literature to the best our knowledge. This question is important because although the nature of criticality is similar as both the spin models lie in the same universality class, there is no guarantee a priori that ANNs would be able to learn the concept of universality automatically during training.

One serious issue in using ANNs in phase detection is that the ANNs may simply learn the value of the magnetization ($\mu$) corresponding to a transition value (for instance, $\mu = 0.5$). While it is remarkable that the ANNs can identify and isolate the order parameter, but the magnetization, being a simple average of spin, by itself does not contain any deep information about the phase transition. To ameliorate the superfluous learning problem, we have introduced an adversarial train-

---

♣ Author names appear alphabetically. All authors contributed equally.
$^{*}$ pallabbasu@gmail.com
$^{†}$ jyoti@phy.iitkgp.ac.in
$^{‡}$ dileeppavansurya@gmail.com
$^{§}$ 956785@students.wits.ac.za
¶ research.vishwanath@gmail.com
$^{1}$ Acronyms: ML: Machine learning, ANN: artificial neural net, CNN: convolutional neural net, DNN: dense neural net; a CNN and a DNN are type of ANNs. Also, $\mathcal{NN}$ : nearest neighbor, $\mathcal{NNN}$ : next to nearest neighbor.

ing procedure [17, 18] by simply shuffling the spins. We denominate the resultant dense neural net (DNNs) as aDNNs. This shuffling destroys any spin-spin correlation but keeps the magnetization invariant. Another issue in using ANNs in phase or criticality detection is that only energy of a given configuration is good enough to infer about the phases. Hence, ANNs, in principle, need only to learn a short range function of the spins for phase detection. However our testing on the wild data sets gives us confidence that aDNNs learned long range spin-spin correlations. To conclude, let us summarize the primary achievements of our aDNN models in detecting the critical region:

1. Our aDNNs perform as good as regular DNNs.

2. Unlike a simple magnetization and energy based classifier, our aDNNs can transfer the learning reasonably well to wild data-sets.

3. Unlike a trained DNN, our aDNN remains blind to the obvious order parameter (magnetization) and is robustly resistant to shuffling attacks.

The paper is organised as follows. As a starting point of our analysis, we provide a physically well motivated definition of *'critical region'* on a finite lattice in §I A.

Before getting in to our main problem of detecting criticality, in §II we start by studying the success of DNN and CNN based classifiers to predict the phases of the 2D Ising model [8]. We also discuss how well a low-dimensional magnetization (and energy) based classifier performs compared to a full-fledged DNN on the entire lattice. To study the ability of our ML models to learn about the phases, we test the performance of ML models on 'wild' datasets with $\mathcal{NNN}$ interactions, after training our models on the Ising model only with $\mathcal{NN}$ interactions (see Fig.1). Here a full-fledged DNN shows a clear improvement over models with low dimensional classifiers. In §II B we also find that the same models gets deceived by an adversarial attack based on shuffled spin configurations. This demonstrates that these models learn mostly about magnetization to identify the phases.

In §III we report on the training and performance of specially designed ANN models to detect criticality. We introduce adversarial training and test the model performance on $\mathcal{NNN}$ datasets. Our models are able to identify the 'critical region' with satisfactory accuracy, simply by examining the individual spin configurations. We also explicitly demonstrate that adversarial training ensures that these models have not learnt anything about magnetization, implying that the success of these models is based on an 'understanding' of spin-spin correlations near criticality. We believe that our training procedure mimicking adversarial attacks can be readily generalized to other physical systems where it can help direct and control the learning procedure.

We conclude with some future outlook in the discussion Section §IV. In Appendix-A we perform similar adversarial training with a CNN, which is not as success-

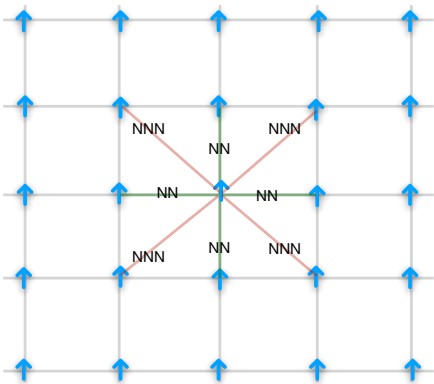

FIG. 1. Ising spins with nearest neighbour ($\mathcal{NN}$ ) and next-to-nearest neighbour $\mathcal{NNN}$ interactions on a two-dimensional lattice. The Hamiltonian is given in (1).

ful as our aDNN. In Appendix-B we study the effect of Fourier transform of the spin configuration data in training and testing our phase predicting models. Finally in Appendix-C we provide some relevant details of our models and their training procedure.

### A. The Ising model: Generating the Data-sets

In this paper, we consider the Ising model on a square lattice with $\mathcal{NNN}$ interactions. The Hamiltonian of the system is given by

$$\mathcal{H} = -J \sum_{<ij>}^{\mathcal{NN}} s_i s_j - g \sum_{<ij>}^{\mathcal{NNN}} s_i s_j, \qquad (1)$$

where $s_i$ is the spin at lattice site $i$, $J$ the $\mathcal{NN}$ coupling constant and $g$ the $\mathcal{NNN}$ coupling constant. The next-to-nearest neighbour spins are the ones which lie diagonally in relation to a given spin (see Fig.1).

We simulate this Ising model on a $40 \times 40$ lattice using Monte Carlo (MC) method. We set the lattice spacing to be unity throughout our analysis. With this choice of units we have to specify three other parameters $J$, $g$ and temperature $T$ in order to carry out the numerical analysis. In this paper, the set of values of the couplings which we consider are $J = 1$; $g = 0$, 0.1, 0.5, 1 and $J = 0.01$, $g = 1$. Corresponding to each of these pair of values, we simulate spin configurations for different range of values of temperature as shown in Table I. For a given set of couplings, the range of the temperature is chosen such that the critical region is contained well within the range. In Fig. 2 and Fig. 3, we plot the magnetization and the susceptibility, respectively, as a function of temperature for different values of the coupling constants. In both these figures the solid dots on the curve denote the values of temperature for which the MC simulation has been performed. The values of temperature have been chosen at an interval of 0.025 in the critical region while

TABLE I. The various values of couplings used in our analysis, temperature range for which the MC simulations were performed, value of $T_c$ as inferred from the maxima of susceptibility (see Fig.3), range of temperature for the critical region.

| Coupling values | Temperature range | $T_c$ [a] | Critical region |
|---|---|---|---|
| $J = 1, \ g = 0$ | $[1.000, 4.000]$ | $[2.300, 2.325]$ | $[2.200, 2.625]$ |
| $J = 1, \ g = 0.1$ | $[1.000, 4.000]$ | $[2.625, 2.650]$ | $[2.525, 2.975]$ |
| $J = 1, \ g = 0.5$ | $[2.500, 5.500]$ | $[3.875, 3.900]$ | $[3.725, 4.350]$ |
| $J = 1, \ g = 1$ | $[2.000, 7.000]$ | $[5.350, 5.375]$ | $[5.100, 6.100]$ |
| $J = 0.01, \ g = 1$ | $[1.000, 4.000]$ | $[2.450, 2.475]$ | $[2.350, 2.825]$ |

[a] The $T_c$ values lies within range of temperature specified in this column. Our MC simulations have been performed for discrete values of temperature with a spacing of 0.025 in the critical region.

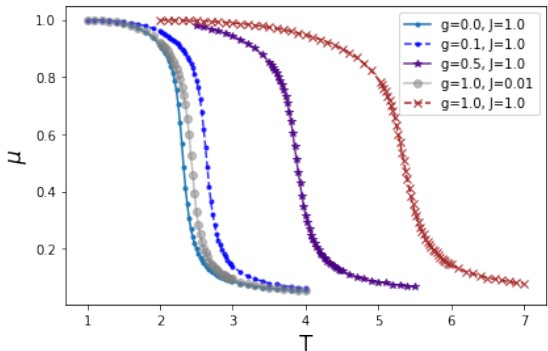

FIG. 2. Magnetization ($\mu$) as a function of temperature for the various values of couplings as evaluated from our data-sets.

in the non-critical region this interval has been increased to 0.1. For every value of temperature, we have accumulated $10^4$ spin configurations which are later used for training and testing our ANNs.

The critical temperature $T_c$, for a given value of $J$ and $g$, is defined as the temperature at which the susceptibility curve exhibits a maxima, see Fig. 3. It may be noted that for $g = 0$ our simulation provides a $T_c$ between 2.3 and 2.4 which differs from the standard value 2.26 obtained by exact analytical methods. This discrepancy is due to the finite size of our lattice. It is of course possible to recover this exact value of $T_c$ by performing finite-size scaling, but since here we use the $40 \times 40$ lattice configurations to train and test our ANNs, we shall consider the value of $T_c$ which may be inferred directly from these finite lattice configurations. Therefore, these values of $T_c$ may be slightly different from their infinite lattice counterparts. This is particularly important since we use these values of $T_c$ to label the ordered and disordered configurations while training and testing our models.

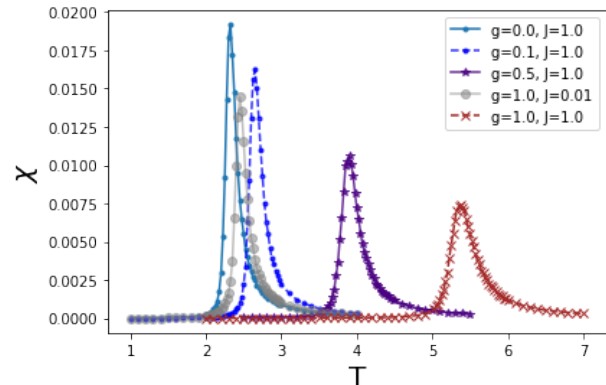

FIG. 3. Susceptibility ($\chi$) as a function of temperature for the various values of couplings as evaluated from our data-sets.

### 1. Critical or Crossover Region

One of the main objectives of our paper, is to demonstrate the ability of our ANNs to learn and identify the critical region besides the ordered and disordered phases. Although we understand the critical region to be the region near the transition point $T_c$, the boundary of this region on the temperature axis is not sharply defined. However, in order to be able to train our ANNs it is important for us to have a precise quantitative definition of the 'critical region'.

One of the most natural method to define the critical region is to use the concept of correlation length. We know that for the 2D Ising model, the spin-spin correlation function near the critical region behaves as

$$G(\mathbf{r}_{ij}) \equiv \langle s_i s_j \rangle = f(r)e^{-r/\xi}, \tag{2}$$

where $f(r)$ has a polynomial fall-off and the correlation length $\xi$ diverges as we approach the critical point in an infinite system. For a finite lattice, as in our case, it remain finite but reaches a maximum value at $T_c$. On the lattice, instead of finding $\xi$ from the asymptotic decay of $G(\mathbf{r}_{ij})$, we may use its Fourier coefficients to arrive at a definition of $\xi$ which is more easy to compute from the spin configurations directly (see the review [16]).

The discreet Fourier transform of the spin-spin two point function $G(\mathbf{r})$ gives the structure factor

$$S(\mathbf{k}) = \sum_{\mathbf{r}} G(\mathbf{r}) \ e^{-i\mathbf{k}\cdot\mathbf{r}} = \langle \sigma_{-\mathbf{k}} \sigma_{\mathbf{k}} \rangle, \tag{3}$$

where

$$\sigma_{\mathbf{k}} = \frac{1}{\sqrt{N}} \sum_{j} s_j \ e^{-i\mathbf{k}\cdot\mathbf{r_j}}, \tag{4}$$

the total number of lattice points being $N$. Now let us consider $\mathbf{k}_1 = (2\pi/L)\hat{k}_x$ and $\mathbf{k}_2 = (4\pi/L)\hat{k}_x$ where $\hat{k}_x$ is the reciprocal lattice unit vector in the $x$-direction and $L = 40$ in units of lattice spacing. With the help of this

structure factor, we can now define a correlation length (see [16]) [2]

$$\xi_b = \frac{L}{2\pi} \sqrt{\frac{S(\mathbf{k}_1)/S(\mathbf{k}_2) - 1}{4 - S(\mathbf{k}_1)/S(\mathbf{k}_2)}}. \tag{5}$$

This $\xi_b$ is an excellent approximation of the correlation length $\xi$ obtained from the asymptotic behaviour of the correlation function near the critical point. As a matter of fact, this quantity is finite and well defined for all range of temperature, even as we approach $T = 0$. We shall therefore, use this quantity for defining the critical region.

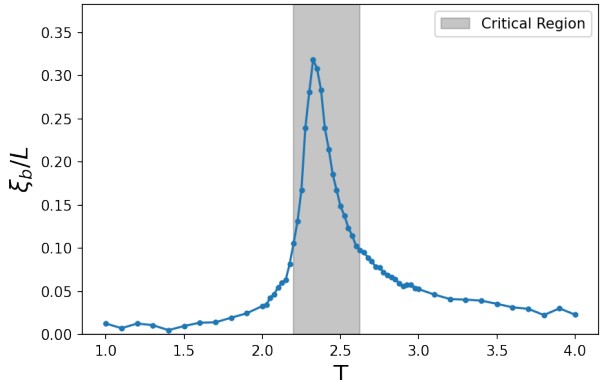

FIG. 4. The plot of $\xi_b/L$ for the $J = 1, g = 0$ data-set. The critical region is chosen according to the condition $\xi_b > 0.3 \, \xi_b^{max}$. The choice of 0.3 is motivated by the temperature range where the change of the order parameter is substantial (see Fig. 2).

In Fig.4 we plot $\xi_b/L$ as a function of temperature for $J = 1, g = 0$ training data set. We see that this $\xi_b$ has a maxima at $T_c$. Let us denote this maximum value by $\xi_b^{max}$. We then define the critical region around $T_c$ as the range of temperature for which

$$\xi_b > 0.3 \, \xi_b^{max}. \tag{6}$$

This is the definition of critical region which we shall use throughout this paper. A plot similar to Fig.4 can be generated for the rest of the data-sets for different values of the coupling constants; the critical region for the individual data-sets obtained through this definition has been reported in Table I. We shall classify the training data-set ($J = 1, g = 0$) into 'critical' and 'non-critical' based on the condition (6). Hence, this choice is somewhat important for training purpose. It is clear that this

---

[2] This definition of correlation length is not unique (see [16] for an alternative definition). But for the purpose of this paper, we will use this definition since it is independent of S(0) (which is simply the square of magnetization); moreover, it remains finite for all values of temperature.

definition of the critical region is not unique (particularly the choice of 0.3), but our main result about the ability of the our model to predict the critical region is not qualitatively affected by this ambiguity.

## II. MACHINE LEARNING OF PHASES

### A. What do machines learn about the phases?

At first let us point out that a given lattice configuration is associated with a temperature $T$ through the probabilities $P(T|\{s_i\})$. Probabilistic nature of the problem is due to the fact that for a finite-size lattice, a given configuration may come from different temperatures with probabilities given by the usual formula for the canonical ensemble,

$$P(\{s_i\}|T) = \exp(-E(\{s_i\})/T)/Z(T), \tag{7}$$

where $Z(T)$ is the partition function at temperature $T$. In an infinite lattice (the thermodynamic limit), it is possible to associate a temperature with a given configuration via its energy. No such unique correspondence exists for a finite system, where temperature is, at best, a property of a collection, not of an individual configuration. From (7) we may infer,

$$P(T|\{s_i\}) = P(T|E(\{s_i\})). \tag{8}$$

In words, the above equation means that details of a spin configuration is somehow irrelevant, and knowing the energy alone is good enough to get a reasonable prediction for temperature. However, either way, a direct computation of $P(T|\{s_i\})$ is hard! We may approximate the $P(T|\{s_i\})$ with the output of a trained ML model as $P_{\mathrm{ML}}(T|\{s_i\})$.

More generally, the discussion in above paragraph applies to the prediction of any kind of temperature based categories like high and low temperature phases. Say, $X$ is a criterion based on temperature labels $T$. For simplicity let us take a binary criterion based on phases. We would train any ML model, by maximizing binary cross entropy between $X$ and the probabilities predicted by the ML model,

$$\mathrm{BCE}(X; P_{\mathrm{ML}}(X|\{s_i\})) \leq \mathrm{BCE}(X; P(X|\{s_i\})) \tag{9}$$
$$= \mathrm{BCE}(X; P(X|E(\{s_i\})))$$

Hence, it is generally expected that an ANN aware of energy would perform well or better than any other predictive model and hence can be used as a benchmark of training. We do not expect the energy based model to perform well when applied to a wild data-set, as energy does not contain any deep information about long range spin-spin correlations.

At first closely following [8], we shall train vanilla ML models (both DNNs and CNNs) to distinguish the phases of the Ising model. In this respect, our most important

TABLE II. Percentage accuracies in prediction of phases by various ANN models on the data-set of spin configurations of the 2D Ising model with nearest neighbour interaction strength $J$ and next-to-nearest neighbour interaction strength $g$.

| Value of $J$ and $g$ | DNN-2 [a] | DNN-4 [b] | CNN | FFT DNN-2 [c] | FFT DNN-4 [d] | FFT CNN [e] | $\mu$ Classifier [f] | $(\mu, E)$ Classifier [g] |
|---|---|---|---|---|---|---|---|---|
| $J = 1, g = 0$ (critical) | 88.3 | 87.7 | 86.3 | 85.8 | 86.0 | 85.9 | 88.2 | 88.7 |
| $J = 1, g = 0$ (non-critical) (tested on training data-set) | 100 | 100 | 100 | 100 | 100 | 100 | 100 | 100 |
| $J = 1, g = 0.1$ (critical) | 86.9 | 86.4 | 88.2 | 85.2 | 85.4 | 84.4 | 85.3 | 86.3 |
| $J = 1, g = 0.1$ (non-critical) | 99.9 | 99.9 | 100 | 100 | 100 | 100 | 100 | 100 |
| $J = 1, g = 0.5$ (critical) | 87.4 | 87.0 | 89.7 | 88.2 | 88.3 | 85.5 | 85.3 | 85.8 |
| $J = 1, g = 0.5$ (non-critical) | 99.9 | 99.9 | 99.9 | 100 | 100 | 100 | 99.9 | 99.9 |
| $J = 1, g = 1$ (critical) | 88.3 | 88.0 | 86.8 | 89.8 | 90.1 | 85.9 | 86.8 | 86.7 |
| $J = 0.01, g = 1$ (non-critical) | 100 | 100 | 99.8 | 100 | 100 | 100 | 100 | 100 |
| $J = 0.01, g = 1$ (critical) | 90.4 | 90.4 | 78.4 | 91.6 | 91.7 | 68.5 | 84.8 | 72.8 |
| $J = 0.01, g = 1$ (non-critical) | 100 | 100 | 97.7 | 100 | 100 | 99.8 | 99.9 | 99.5 |

[a] Shallow fully connected DNN with 2-layers.
[b] Deeper fully connected DNN with 4-layers.
[c] Fully connected 2-layers DNN trained with Fourier transform (only magnitude) of spin-configuration data (see Appendix-B).
[d] Fully connected 4-layers DNN trained with Fourier transform (only magnitude) of spin-configuration data (see Appendix-B).
[e] CNN trained with Fourier transform (only magnitude) of spin-configuration data.(see Appendix-B)
[f] DNN based simple classifier trained on magnetization only.
[g] DNN based simple classifier trained on magnetization and energy only.

addition to [8] is that we train our models on the usual $\mathcal{NN}$ ($J = 1, g = 0$) dataset, *away from the critical region* and also test the performance of our model on a completely new data set (wild data) characterized by a different value of $g$. This important modification tests the ability of our models to learn the universal features of the phase transition. For the bench-marking purpose we also train two simple classifiers, one aware only of magnetization and another with energy included (also see [14]). This reduces the data associated with a spin configurations from $40 \times 40 = 1600$ dimensions to one (or two) dimensional subspace.

We shall now report on the accuracy of our binary classifier models in identifying phases of the 2D Ising model. Details of the model architectures and training is presented in the Appendix C. It is worth pointing out that the training of our models only in the non-critical region is more of a necessity than a preference. This is because, if we train our models at temperatures very close to the critical point, the models get confused and the learning is sub-optimal. Consequently, the performance of the models trained very close to $T_c$ are highly compromised.

The prediction accuracies of our models has been detailed in Table II separately for the critical and the non-critical regions. For the non-wild data, the accuracies are close to the 'maximal accuracy' of $(\mu, E)$-classifier. As expected the accuracies are lower in the critical region [3] compared to non-critical temperatures, where the success of our models is outstanding. Here we use the same definition of critical region as presented in Table I.

Note that for $g = 0$ the critical region data is a

________

[3] The reader may note that our accuracies in the critical region is comparable to [8]. Note that we consider temperature values at a spacing of 0.025 compared to 0.25 used in [8]. Hence, although we have more points closer to the critical temperature our overall accuracies in the critical region are not significantly affected.

part of our 'test' data since we do not train our models here. We also test our model for wild data-sets $J = 1$, $g = 0.1$, $0.5$, $1$ and $J = 0.01$, $g = 1$ to check the transferability of learning to other forms of interactions within the same universality class. Interestingly, we find that the accuracy of the detection of phases is practically unaffected by the introduction of non-zero $g$. In fact there is also no appreciable change in the performance of our models on the $J = 0.01$, $g = 1$ data-set. We believe that this robustness of the predictive power of our models within the same universality class may be an extremely useful observation for future application of ANNs to address similar questions. However a drop in accuracy is observed for the simple classifier with energy. Unlike magnetization, energy of configurations near phase transition change drastically as we change the value of the couplings, and accordingly energy become a bad predictor.

## B. Adversarial attack: Fooling a trained ANN with shuffled spins

The relatively high success of our models in Table II may be slightly misleading. ANNs outperform a simple magnetization/energy based classifier, although the differences are not huge. In four out of five examples (see Table II) a simple $\mu$ classifier perform only about 2% less than ANNs. As we have mentioned in §I, a criticism of machine learning of phases could be that ANNs too had secretly learned about the order parameter (magnetisation) with large weightage while giving any other feature related to the physics of spin-correlations a small to negligible weight. Note that magnetization is just an average of all the spins in a given configuration and as such could be calculated by one linear hidden layer with equal weights! Hence no complicated ANN machinery is at all needed and phase detection from order parameter seems to be a superficially trivial problem.

To investigate the role of magnetization, we designed a simple adversarial attack. Instead of testing our trained model directly on the data-set, we randomly shuffle the spins. This ensures that the magnetisation of every configuration is intact, but the spin-spin correlations are all destroyed. The individual spin configuration are labeled 'ordered' or 'disordered' using the corresponding temperatures, borrowing the value of $T_c$ from the original configurations prior to shuffling. As a result magnetization is the only reliable data in these spin configurations which is consistent with this labeling, all other spin-correlations being lost in the process of shuffling. The result of testing our model on these non-physical configuration is reported in Table III.

We find that there is a some decrease in the prediction accuracy, but still our models are deceived to predicting the correct phases to a great extent. The fact that there is a drop in the predicting accuracy suggest that the model learns more than just the magnetization. However, the

TABLE III. Phase prediction accuracies on data with shuffled spins (adversarial attack). The models have been trained using non-critical data for the $J = 1, g = 0$ data-set (unshuffled). They are the same models which have been used in the first three columns of Table II. Note that although the shuffled data is junk (except the information of magnetization) yet the phase prediction accuracies are comparable to the first three columns of Table II.

| Coupling values | DNN-2 | DNN-4 | CNN |
|---|---|---|---|
| $J = 1$, $g = 0$ (critical) | 86.3 | 87.3 | 80.8 |
| $J = 1$, $g = 0$ (non-critical) | 100 | 100 | 98.0 |
| $J = 1$, $g = 0.1$ (critical) | 84.8 | 85.9 | 79.4 |
| $J = 1$, $g = 0.1$ (non-critical) | 99.9 | 99.9 | 95.7 |
| $J = 1$, $g = 0.5$ (critical) | 85.6 | 86.6 | 74.7 |
| $J = 1$, $g = 0.5$ (non-critical) | 99.9 | 99.9 | 92.2 |
| $J = 1$, $g = 1$ (critical) | 86.8 | 87.6 | 73.3 |
| $J = 1$, $g = 1$ (non-critical) | 100 | 100 | 96.0 |
| $J = 0.01$, $g = 0.1$ (critical) | 90.0 | 90.3 | 76.1 |
| $J = 1$, $g = 0.1$ (non-critical) | 100 | 100 | 94.4 |

fact that it is still able to predict phases on junk data (with randomised spins), also suggests that magnetization is the most important feature in the learning process, and it is very likely that a high relative weight is assigned into learning magnetization. It is also noteworthy that the drop in accuracies is more for the CNN models than for the connected networks; this suggests that the former is able to learn some additional aspects of a phase besides the order parameter.

## III. MACHINE LEARNING OF CRITICALITY

We can now conclude that the ANN training algorithms are able to learn the notion of order parameter well. Therefore, the existence of such an order parameter makes the process of learning about the phases, in some sense, trivial. Hence, we would now like to investigate whether our algorithms are able to learn something deeper about the spin configurations. This motivates us to investigate the following question. Is it possible for the ANNs to learn about the 'critical region' instead of identifying just the phase? In this section, we shall report significant progress on this question.

Let us keep in mind that we are dealing with a finite-size lattice. Actually in the phase space, we do see a lot

of overlap between quantities calculated at different temperature. Hence there is a limit on how accurately we may classify the spin configurations based on any temperature/magnetization based categories. For instance see the overlaps in Fig. 5. Due to these overlaps the best possible prediction accuracies are less than 100%.

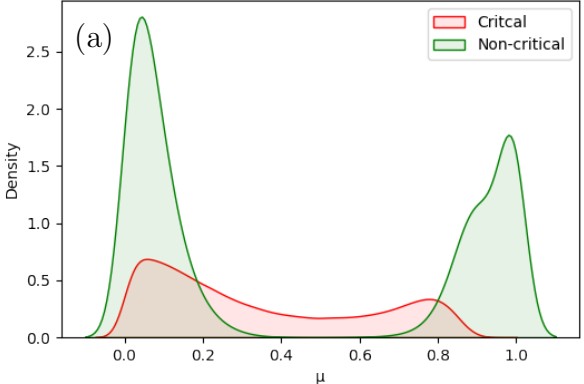

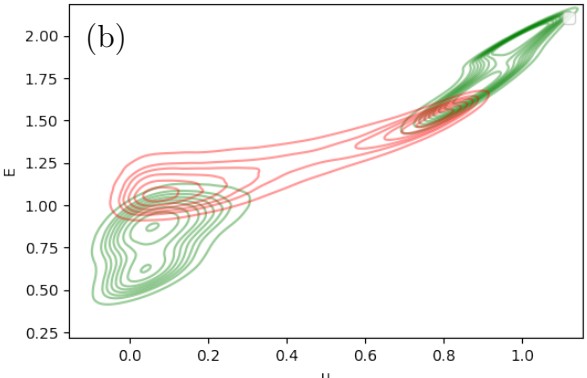

FIG. 5. How the density of the critical and non-critical configurations overlap in the $\mu$ and $(\mu, E)$ subspace. These overlaps puts an inherent limit on classification.

## A. Identifying Criticality: Training and testing DNNs on the $\mathcal{NN}$ data set

We shall train and test DNNs to identify the critical spin configurations. For a given value of $J$ and $g$ we shall classify the temperatures as critical or non-critical based on the definition of §I A 1 (see Table I). Here we begin by dividing the $J = 1$ and $g = 0$ data-set into training (80%) and testing (20%) data-sets. We then train a 4-layer DNN to identify the critical region and subsequently test its ability to predict the same in the test data-set. The details of our architecture and training are described in the Appendix C.

From the predictions we accumulate the probability, which lies between 0 and 1. We average this quantity

for all the configurations that correspond to a given temperature. In Fig. 6 we show a comparison of the performance of various models; in particular, Fig. 6c (solid green line) is for a 4-layer DNN without adversarial training (DNN-4). For visual comparison, we mark all the plots with critical region predicted from the correlation length. We conclude that the trained model is able to identify the critical region very well. Just as in the prediction of phases II. For comparison, we also train two low-dimensional classifiers II A: One with only magnetization ($\mu$ ) and another with energy (and $\mu, E$) included, see Fig. 6 (a) and (b). To the credit of ML, we notice that a magnetization based classifier works worse than a full-fledged DNN/CNN trained on the entire lattice. The DNN based model has a performance close to the energy based model.

## B. aDNN: Adversarial training with shuffling

We shall now employ a suitably designed adversarial training [19] to force our models to ignore magnetization during the learning process. This is achieved in the following way. During training, apart from the original spin configurations, we will also use non-physical configurations where the spins are randomly shuffled, labeling them as non-critical even for temperatures within the critical region. As usual, the random shuffling will eliminate all spin-spin correlation keeping the magnetization intact. Consequently, the training data-set now contains spin configurations which can be critical or non-critical for the same value of magnetization. This ensures that magnetization is definitely not a feature which is learned by our models to identify criticality. So, now the question remains, whether a model trained in this fashion can still identify the critical region in the original (unshuffled) test data-sets.

As in most cases of training in this work, we train the models with 80% of the $J = 1, g = 0$ data-set. The remaining 20% constitutes the test data-set. These models, now equipped with adversarial training, are then tested on $J = 1, g = 0$ test data set (both original and shuffled). We show the results of this test in Fig. 6(d). The fact that the magenta curve is close to zero, while the solid green curve is almost unaffected indicates that our adversarial training has been successful. However, we observe that CNNs seem to perform worse with adversarial training, see Appendix A and Fig. 8.

## C. Testing in $\mathcal{NNN}$ wild datasets with non-zero $g$

We then go on to test our DNN and aDNN models on data-sets (both original and shuffled) corresponding to a non-zero $g$. In Fig. 7 we display our test results on these wild data-sets, which clearly show that our model identifies the shuffled spin configurations as non-critical while maintaining a good accuracy in identifying the critical

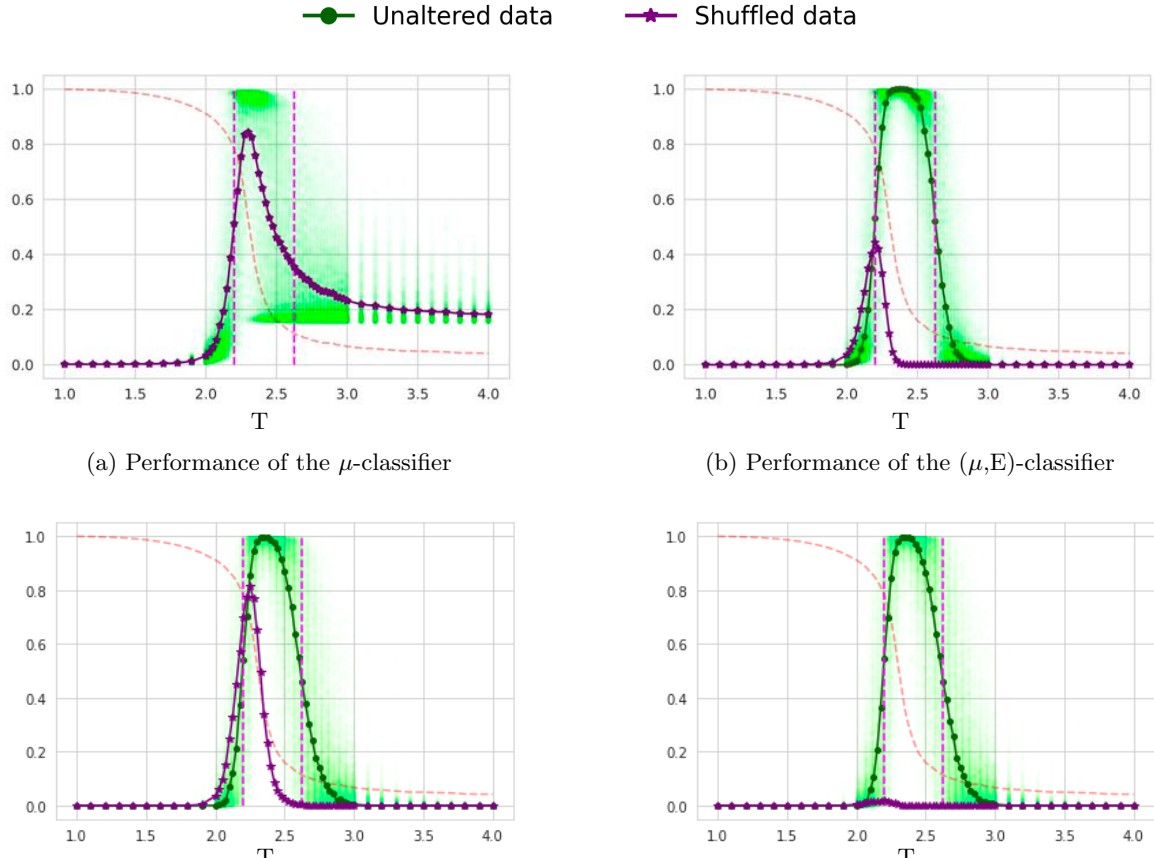

(a) Performance of the $\mu$-classifier

(b) Performance of the $(\mu,E)$-classifier

(c) Performance of the DNN-4 model *without* adversarial training.

(d) Performance of the aDNN-4 model *with* adversarial training.

FIG. 6. Comparison of the performance of various models to predict the critical region for the test data corresponding to $J = 1$, $g = 0$. In (a) and (b) we show the performance of simple classifiers trained on only $\mu$ and $(E, \mu)$ respectively. In (c) and (d) we report the performance of a 4-layer DNN without(DNN-4) and with (aDNN-4) adversarial training. The bold green curve represents the performance of the models on unaltered test data. The magenta curve represents the performance of the models on unphysical data (adversarial attack) where the spins are randomly shuffled in a configuration. In (a) there is no difference between the green and the magenta curves since the magnetization ($\mu$) does not change with spin-shuffling. The probability for every spin configuration to be critical has been shown with the light-green background; the dark green curve is the average of all the light-green values for a given temperature. The magnetization (red) and critical region (pink) are shown with dashed lines in the background. Since the shuffled data has no critical region in terms of the spin-spin correlations, the magenta line is expected to be zero everywhere for a model which has learnt the physical notion of criticality. Clearly, the aDNN-4 model with adversarial training, trained on the full spin-configurations, has the best performance.

region in the original data. Simple classifiers based on energy and magnetization understandably perform worse in identifying critical region in the wild data.

Note that we have chosen to represent the performance of our models through the plots in Fig. 7 displaying probability (for all the individual tested configurations, see the light-green scatter). This representation is advantageous since we do not need to demarcate the critical region a priori for the wild ($g \neq 0$) data-sets. Therefore, a plot like this carries more information than something like the average accuracy percentage against a dubious pre-defined label.

To conclude this section, let us summarize the primary achievements of our aDNN model (DNNs with adversar-

ial training):

1. Our aDNN performs close to a 'maximally accurate' energy based classifier for the $g = 0$ test data-set.

2. Unlike a simple magnetization and energy based classifier, our aDNNs can transfer the learning reasonably well to the wild data-sets with $g \neq 0$.

3. We are also able to clearly demonstrate that our aDNN remains blind to the obvious order parameter (magnetization) and are robustly resistant to shuffling attacks.

From these observations we may logically speculate that

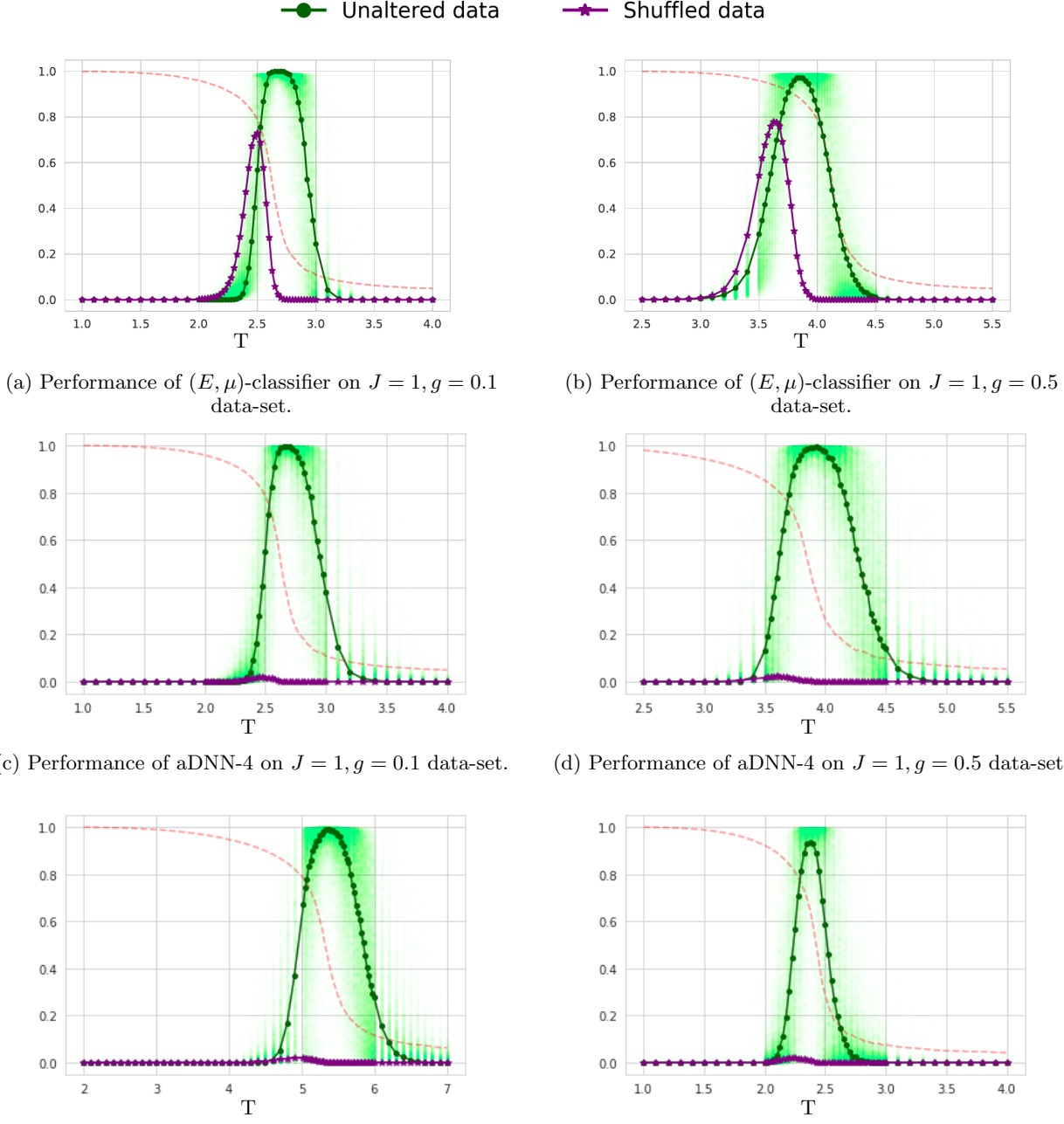

(a) Performance of $(E, \mu)$-classifier on $J = 1, g = 0.1$ data-set.

(b) Performance of $(E, \mu)$-classifier on $J = 1, g = 0.5$ data-set.

(c) Performance of aDNN-4 on $J = 1, g = 0.1$ data-set.

(d) Performance of aDNN-4 on $J = 1, g = 0.5$ data-set.

(e) Performance of aDNN-4 on $J = 1, g = 1$ data-set.

(f) Performance of aDNN-4 on $J = 0.01, g = 1$ data-set.

FIG. 7. Here we test the ability of our models which, have been trained on the $J = 1$, $g = 0$ data-set, to predict the critical region in the 'wild' data-set with next-to-nearest neighbour coupling turned on ($g \neq 0$). In (a) and (b) the performance of the simple classifier trained on $(E, \mu)$ has been presented; they have no adversarial training. In (c), (d), (e) and (f) the performance of the aDNN-4 model (a 4-layer DNN augmented with adversarial training) has been presented. In this figure, we follow the same conventions of plotting as in Fig. 6. The prediction of the critical region by our models (dark green line) in these 'wild' data-sets is compared with region of temperature over which the average magnetization (red dashed line) changes from 0 to 1. It is remarkable that our aDNN-4 model is able to correctly predict the critical region in the unaltered 'wild' data-sets, while predicting absolutely no critical region for the corresponding shuffled data-sets (magenta line).

our aDNN has acquired the ability to identify criticality by learning a non-trivial concept like long range spin-spin correlations.

## IV. DISCUSSIONS

In this paper, we have demonstrated that ANNs are capable of learning additional features of critical behaviour

rather than just the order parameter. Through a series of ML experiments on the spin configurations of the 2D Ising model generated by Monte Carlo methods, we have shown that deep learning models are able to recognize near critical temperatures without learning about magnetization. This implies that, when suitably trained, ANNs can learn spin-spin correlations very well.

At first, we have focused on the ability of ANNs to correctly identify the phases of the 2D Ising model. In this regard, we have clearly established that the learning of our ANNs is transferable to similar spin systems within the same universality class but with a different nature of interaction. Thereafter, we have observed that the success of the ANNs to predict the phases is largely based on learning the value of the order parameter (magnetization) with maximum weight. To show this, we have used unphysical spin configurations where the spins are randomly shuffled thus destroying all correlations, but keeping the magnetization unchanged. We find that even for such unphysical configurations, our models are able to predict the phases with reasonable accuracy. This is only possible if magnetization is the main feature learned by these models, since this is the only property in the unphysical configurations which is unaffected by randomization. We identify this exercise as a form of adversarial attack which has been successfully employed in other applications of machine learning [18].

We then went on to investigate whether our ML models are able to identify the range of temperature for which the Ising model was near critical. We have achieved a significant success in this regard. Further, through a suitably established procedure of adversarial training we were able to guide our models to identify criticality by understanding spin-spin correlations, rather than using magnetization. We believe most of our observations reported here would be easily generalizable to other spin models beyond the 2D Ising model, most of which seems machine learnable [20–23].

As an immediate extension of our work, it would be interesting to investigate whether we can use the concept of adversarial training to enable our models to identify the phases without the knowledge of magnetization. It may be achieved by training a multi-class classifier to identify critical-ordered, critical-disordered and non-critical configurations, with the shuffled spin configurations labeled as non-critical. It would be very interesting to compare the performance of such a classifier with the binary classifier used in this paper to identify the phases within the critical region. We leave this question for future investigation.

We find it noteworthy that our models based on a CNN architecture does not perform as well as the fully connected deep networks when we change the nature of the interactions. It is possible that since the CNNs are known to posses an enhanced ability to identify the local properties of the spin configurations, they find it relatively difficult to learn the universal features of the critical region. It would be very interesting to verify this specu-

lation and find an appropriate method to mitigate this shortcoming.

In a recent paper [24], it has been demonstrated that variational autoencoders with a novel architecture are able to identify the critical region of the 2D Ising model very well. It would be very interesting to test such self-supervised learning with the adversarial attack discussed in this paper. In general, it would be very interesting to decipher which aspect of the criticality are learnt by these autoencoders.

Also, remarkable similarities of the structure of CNNs with renormalization group flows have been highlighted in the recent literature [3, 25–29]. Since the $\mathcal{NNN}$ interactions are irrelevant deformation of the Ising critical point, a study of the performance of CNNs as we vary the $\mathcal{NNN}$ interaction strength is also significant in this context.

Finally, it would be extremely interesting to explore which other aspects of phase transition or cross-over in spin-models can be reliably learned by multi-class classifiers based on deep networks. For example, given some spin configuration data would it be possible for a ML model to estimate the temperature, correlation length, and the microscopic nature of inter-spin interactions [30]. Another set of extremely important physical quantity associated with phase transitions are the critical exponents. The efficacy of ML models to estimate the critical exponents (see [31] for recent work) may compliment and augment existing tools to compute them.

## ACKNOWLEDGMENTS

We would like to thank Debasish Banerjee for initial collaboration, many useful discussions and for sharing with us the first version of the data generation code where Monte-Carlo algorithm was used for simulating the 2D Ising model with nearest neighbour interactions. We would also like to thank Bonny Banerjee, Vishnu Jejjala, Robert De-Melo Koch, Kannabiran Seshasayanan, Jonathan Shock and Zaid Zaz for many useful discussion. We thank *Kaggle*, particularly for providing access to GPU accelerators, where some of our codes have been tested. JB and VS would like to acknowledge support from the Institute Scheme for Innovative Research and Development (ISIRD), IIT Kharagpur, Grant Nos. IIT/SRIC/PH/RFL/2021-2022/091 and IIT/SRIC/ISIRD/2021-2022/03, respectively. VS acknowledges support from the Start-up Research Grant No. SRG/2020/000993 from SCIENCE & ENGINEERING RESEARCH BOARD (SERB), India. VS also acknowledges National Supercomputing Mission (NSM) for providing computing resources of 'PARAM Shakti' at IIT Kharagpur, which is implemented by C-DAC and supported by the Ministry of Electronics and Information Technology (MeitY) and Department of Science and Technology (DST), Government of India, along with the following NSM Grant DST/NSM/R&D HPC Applica-

tions/2021/03.21.

## Appendix A: Critical region prediction with CNN

In this appendix, we shall report on our attempt to train a CNN based classifier to identify the critical region, following our discussion in §III. Our classifier has two consecutive CNN layers with kernal size 3. It has been trained over 500 epochs and the learning rate has been adjusted using a scheduler. The behaviour of the training loss function has been reported in Fig. 9.

Once trained, we test our model on the 20% test data for $J = 1, g = 0$. The result of this testing has been reported in Fig. 8a. Here, we find a very reasonable performance almost comparable to that of 4-layer DNN in Fig. 6d. In the rest of the plots in Fig. 8 we report the performance of the same model on $J = 1, g = 0.1, 0.5, 1.0$. We see that the performance gradually decreases with increasing value of $g$. In fact, we find that for the data-set corresponding to $J = 0.01, g = 1$, this model is not able to identify any critical region at all.

We would like to speculate that this inferior performance of the CNN can be attributed to the fact that model learns the local features of the spin configurations so well that it has some problems in identifying the universal features in wild-data ($g \neq 0$).

## Appendix B: Enters FFT

We shall now perform a *Fourier transform* (FFT [32]) of our spin configuration and then train and test our models on this Fourier transformed data. However, instead of keeping track of both the real and imaginary parts of the Fourier modes, we will only use the absolute value for the training and testing purposes. With this, we definitely loose some information about the spin configurations, but the question is, how does this affect the prediction accuracies of our models.

Again, we follow the same protocol as before. We train our model for $g = 0$ away from criticality and test our

model in the critical region for $g = 0$, and also on the entire dataset for $g = 0.1, 0.5, 1$ and $J = 0.01, g = 1$. Our results in this case has been reported in Table II. We find that our results are very close to the accuracies of the models trained on the original data. In fact, in several instances, the accuracies involving FFT increases slightly. [4]

We would like to speculate that if we use Fourier transformed data for training, then our models tend to learn the notion of magnetization more easily. This is because the value of magnetization is directly presented to the models during training as the zero mode. We have observed that if we retain only the first few Fourier modes for training and testing our models, we still get a reasonable accuracies of prediction. This observation, in a way, corroborates our speculation.

## Appendix C: Technical details of our models and their training

In the work reported in this paper, we have used the PyTorch framework to train and test our models. We have trained three different models for phase identification in §II A. They constitute two fully connected deep networks and one CNN. Both the Deep networks has a ReLU activation function. The main difference between the two fully connected networks is that one of them has a single hidden layer - a shallow DNN, while the other one has 3 hidden layers. The CNN model used in §II A has a single convolution layer and its kernel size is 2. All three models have been trained over 50 epochs. The evolution of the training loss-function with number of epochs have been reported in fig.10. The performance of this model on original data-sets have been reported in table-II, while that on unphysical shuffled data-sets (adversarial attack) have been reported in Table III.

The DNN model used for the prediction of critical region in §III has a structure very similar to the one used for phase prediction. But in this case, we train the model for 500 epochs. The evolution of the training loss-function in this case, has been reported in Fig. 11.

---

[4] We have observed that sometimes if less training data is available (i.e. the spacing between the temperatures are larger), then training and testing using Fourier transforms in this manner can significantly enhance the accuracies.

[1] G. Carleo, I. Cirac, K. Cranmer, L. Daudet, M. Schuld, N. Tishby, L. Vogt-Maranto, and L. Zdeborová, Reviews of Modern Physics **91** (2019), 10.1103/revmodphys.91.045002.

[2] A. Morningstar and R. G. Melko, arXiv preprint arXiv:1708.04622 (2017).

[3] E. de Melllo Koch, A. de Mello Koch, N. Kastanos, and L. Cheng, arXiv e-prints , arXiv:2002.02664 (2020), arXiv:2002.02664 [cs.LG].

[4] D. H. Ackley, G. E. Hinton, and T. J. Sejnowski, Cognitive Science **9**, 147 (1985).

[5] G. Torlai and R. G. Melko, Physical Review B **94** (2016), 10.1103/physrevb.94.165134.

[6] J. Carrasquilla and R. G. Melko, Nature Physics **13**, 431 (2017).

[7] J. Carrasquilla and R. G. Melko, Nature Physics **13**, 431 (2017), arXiv:1605.01735 [cond-mat.str-el].

[8] P. Mehta, M. Bukov, C.-H. Wang, A. G. Day, C. Richardson, C. K. Fisher, and D. J. Schwab, Physics Reports

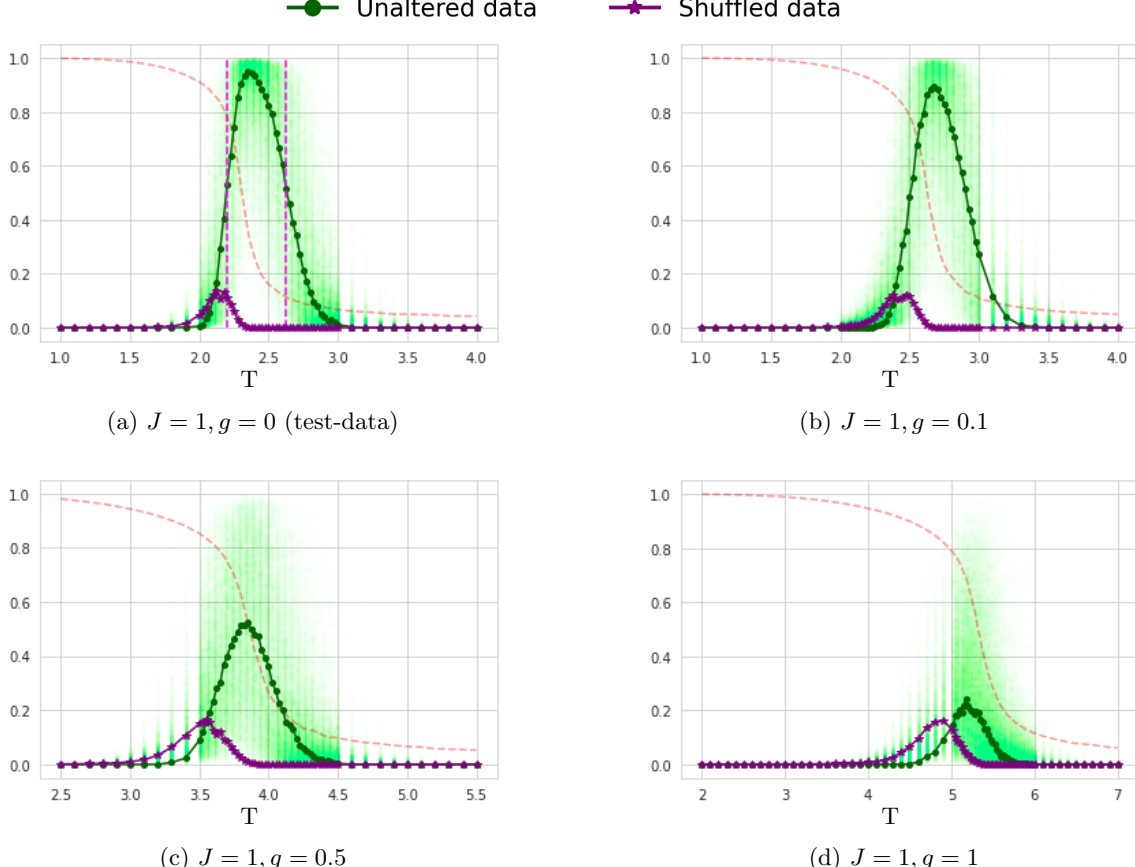

(a) $J = 1, g = 0$ (test-data)

(b) $J = 1, g = 0.1$

(c) $J = 1, g = 0.5$

(d) $J = 1, g = 1$

FIG. 8. The performance of our CNN based model with adversarial training to identify the critical region in various data-sets. We follow the same conventions for plotting as in Fig. 6.

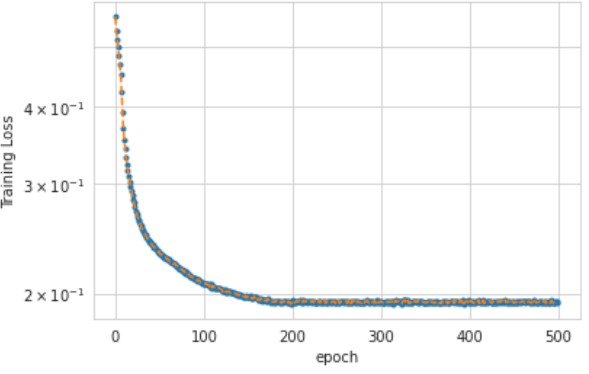

FIG. 9. Training Loss for 'critical region' identification by CNN. The performance of this model has been reported in Fig. 8.

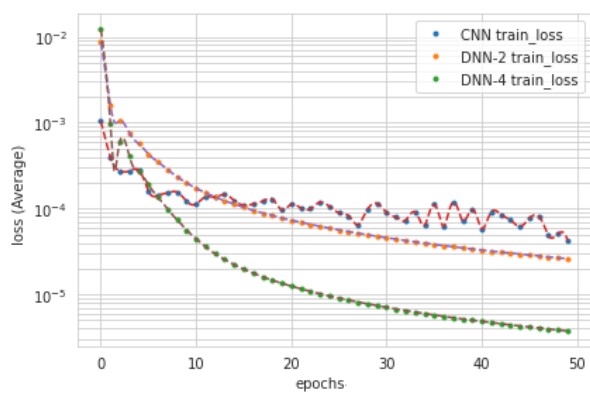

FIG. 10. Average training loss for various models trained to identify the ordered or disordered phase (without FFT).

**810**, 1–124 (2019).

[9] G. Cossu, L. Del Debbio, T. Giani, A. Khamseh, and M. Wilson, Physical Review B **100** (2019), 10.1103/physrevb.100.064304.

[10] S. Efthymiou, M. J. S. Beach, and R. G. Melko, arXiv e-prints , arXiv:1810.02372 (2018), arXiv:1810.02372 [cond-mat.stat-mech].

[11] A. Morningstar and R. G. Melko, arXiv e-prints , arXiv:1708.04622 (2017), arXiv:1708.04622 [cond-mat.dis-nn].

[12] E. De Mello Koch, R. De Mello Koch, and L. Cheng, IEEE Access **8**, 106487 (2020).

[13] S. S. Funai and D. Giataganas, Physical Review Research **2** (2020), 10.1103/physrevresearch.2.033415.

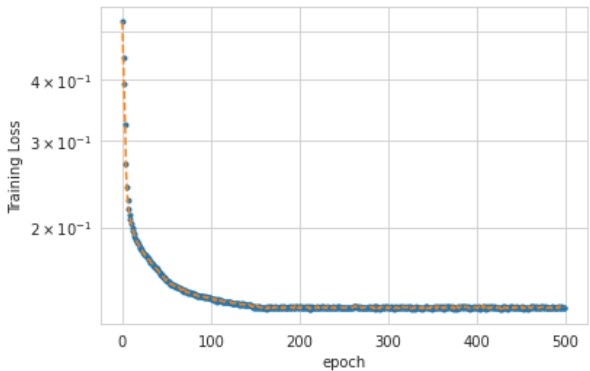

FIG. 11. Training Loss for 'critical region' identification by DNN.

ness," (2021), arXiv:2102.01356 [cs.LG].

[20] W. Bialek, S. E. Palmer, and D. J. Schwab, "What makes it possible to learn probability distributions in the natural world?" (2021), arXiv:2008.12279 [cond-mat.stat-mech].

[21] C.-D. Li, D.-R. Tan, and F.-J. Jiang, Annals Phys. **391**, 312 (2018), arXiv:1703.02369 [cond-mat.dis-nn].

[22] K. Shiina, H. Mori, Y. Okabe, and H. K. Lee, Scientific Reports **10**, 2177 (2020), arXiv:2001.03989 [cond-mat.stat-mech].

[23] D. Giataganas, C.-Y. Huang, and F.-L. Lin, New J. Phys. **24**, 043040 (2022), arXiv:2102.05219 [cond-mat.dis-nn].

[24] N. Walker, K.-M. Tam, and M. Jarrell, Scientific Reports **10** (2020), 10.1038/s41598-020-69848-5.

[25] C. Bény, arXiv e-prints , arXiv:1301.3124 (2013), arXiv:1301.3124 [quant-ph].

[26] P. Mehta and D. J. Schwab, arXiv e-prints , arXiv:1410.3831 (2014), arXiv:1410.3831 [stat.ML].

[27] M. Koch-Janusz and Z. Ringel, Nature Phys. **14**, 578 (2018), arXiv:1704.06279 [cond-mat.dis-nn].

[28] S. Iso, S. Shiba, and S. Yokoo, Phys. Rev. E **97**, 053304 (2018), arXiv:1801.07172 [hep-th].

[29] J. Erdmenger, K. T. Grosvenor, and R. Jefferson, (2021), arXiv:2107.06898 [hep-th].

[30] H. W. Lin and M. Tegmark, Entropy **19** (2017), 10.3390/e19070299.

[31] Z. Li, M. Luo, and X. Wan, Physical Review B **99** (2019), 10.1103/physrevb.99.075418.

[32] Z. Li, N. Kovachki, K. Azizzadenesheli, B. Liu, K. Bhattacharya, A. Stuart, and A. Anandkumar, "Fourier neural operator for parametric partial differential equations," (2021), arXiv:2010.08895 [cs.LG].

[14] D. Kim and D.-H. Kim, Phys. Rev. E **98**, 022138 (2018).

[15] H.-Y. Chen, Y.-H. He, S. Lal, and M. Z. Zaz, "Machine learning etudes in conformal field theories," (2020), arXiv:2006.16114 [hep-th].

[16] A. W. Sandvik, AIP Conf. Proc. **1297**, 135 (2010), arXiv:1101.3281 [cond-mat.str-el].

[17] N. Liu and P. Wittek, Physical Review A **101** (2020), 10.1103/physreva.101.062331.

[18] S. Jiang, S. Lu, and D.-L. Deng, "Adversarial machine learning phases of matter," (2021), arXiv:1910.13453 [cond-mat.dis-nn].

[19] T. Bai, J. Luo, J. Zhao, B. Wen, and Q. Wang, "Recent advances in adversarial training for adversarial robust-