# Peer review of "Machine learning of Ising criticality with spin-shuffling"

_SciPost Physics Core_

## Round 2 · Referee Report · Anonymous (Referee 1) · 2023-3-20

Strengths

Figures 6 and 7 are the most relevant results of the paper. It is a well designed and well performed numerical experiment to produce a network that detects the (arbitrarily defined) critical region without learning the magnetization nor the energy.

The data seems of high quality, according to Figs 2, 3 and 4. Usually is difficult to compute a smooth correlation length for small system sizes (40 x 40 here). The $\mu$-, $E$-, and $(\mu,E)$-classifiers were a very helpful comparison.

Weaknesses

Figure 5. requires some additional information and explanations.
The transfer learning results from a NN model to a NNN model are not a novel result. I have mentioned below a reference that studied this issue extensively in similar Ising models.
Furthermore, section II.A is incorrect and/or really not clear and needs to be rewritten.

Report

I believe that, after addressing all the requested changes, the paper will be suitable for publication. The results in figure 6 and 7 are very interesting and they will be valuable for the reader.

Furthermore, as a general acceptance criteria of this journal, authors must provide

1- a link to a repository showing the code they used 2- a link to a repository with the data sets they used

For 1- a common option is GitHub, and for 2- they can use https://data.mendeley.com/, which is free.

Requested changes

Page 1:

1- rephrase "to imbibe the the spirit of universality", which is to informal (and has a typo).

2- The transfer learning procedure of training a network with NN-data and testing the same network with NNN-data was studied extensively in

https://doi.org/10.1016/j.commatsci.2021.110702

in which the authors studied the square lattice, and also the honeycomb lattice.

Also, in this reference https://doi.org/10.1103/PhysRevX.7.031038 Authors did transfer learning changing filling in a Hubbard model.

Please, add these two citations.

Additionally, since you are doing transfer learning without modifying any of the trained weights, it would be nice if you add this reference called

"Optimal transfer protocol by incremental layer defrosting"

https://arxiv.org/abs/2303.01429

mentioning other possibilities.

Page 2:

  • Please abandon the nomenclature "wild datasets". It is not widely used, and it is not formal. A possible replacement could be "NNN-test set". Another possibility would be "test set 1,2,3,..."

Page 3:

  • Please send Fig. 2 and Fig. 3 to an appendix, since they do not contain critical information.

  • Please remove the phrase "Hence, this choice is somewhat important for training purpose."

  • The two-point correlation function in equation (2) is the correct function whenever you have a finite correlation length, i.e., for $T \neq T_c$. Please rephrase the expression "near the critical region" as the necessary scenario for equation(2).

Page 4:

  • Binary cross entropy is minimized when used as an objective (loss) function, not maximized. As it is the negative log-likelihood function in a binary classification, minimizing the binary cross entropy is equivalent to maximizing the likelihood function of the model.

In inequality (9) are you just saying that the ML model will always perform equal or worst than the true probability distribution P(X|{s_i}) (that we don't know) in a finite system? In this case, given the previous discussion, the inequality sign should be reversed.

Page 7:

  • Please specify explicitly which data set are you plotting and mention explicitly that you are computing an histogram, so the y-axis is a probability density.

Also, how is it that $\mu$ can be higher than 1 ?

Page 8:

  • The top left panel does not have the mean value dark green dots. Please add them.

Page 9:

  • Please add the vertical dashed lines showing the critical region in Fig. 7. They were very helpful in Fig. 6.

Page 10:

  • (related to last paragraph of Sec IV) In 10.1088/1742-6596/2207/1/012058 authors show how to use a network to estimate the correlation length, energy and magnetisation from Ising spin configurations doing regression.

Appendix A:

  • How many kernels did each layer have?

Appendix C:

  • How many kernels did the CNN have?
    Note: Kernel size of 2 is extremely uncommon. Standard is 3, and occasionally can be chosen bigger. See for example resnet architecture https://arxiv.org/pdf/1512.03385.pdf

//----------------------------------------------------------------------//

Typos:

  • Page 1:

  • "In recent year, [...]"

  • "While it is remarkable that the ANNs can identify and isolate the order parameter, but the magnetization, being a simple average of spin, [...]"

Page 2:

  • "We denominate the resultant dense neural net (DNNs) [...]"

  • "In this paper, we consider the Ising model on a square lattice with N N N interactions." NN and NNN interactions.

Page 3:

  • " it remain"

Page 7:

  • "calculated at different temperature"

  • "These overlaps puts an inherent limit on classification."

-In upper panel of Fig. 5, "critcal"

Appendix C:

  • "Both the Deep networks has"

---

## Round 2 · Referee Report · Anonymous (Referee 2) · 2023-3-21

Strengths

The application of artificial intelligence tools to the study of phase classification near the critical region is a topic of current interest

Weaknesses

Transfer learning has been used before, but the authors claim to be the first to study it.

Some sections need to be rewritten, they contain many errors that make it doubtful whether what they did is correct and more detailed explanations are missing.

Report

In the paper "Machine learning of Ising criticality with spin-shuffling" the authors studied the performance of neural networks in identifying critical behavior in the 2D Ising model. The authors train DNN and CNN to identify the critical region in the presence of next-to-nearest neighbor interactions. The authors discuss whether the models can identify criticality in contrast to learning the value of magnetization. The authors also implement an adversarial training process to ignore learning about the order parameter while training. The results presented in the manuscript may be interesting. However, the manuscript is not well written and does not meet the journal criterium.

I have several comments that the authors must answer before I consider recommending publication.

The authors say "we choose the N N data set to train and N N N data sets to test the performance of the ANNs. Although a lot of work has been already done on the prediction of phases in the Ising model, this investigation in the presence of NNN interactions has never been reported in the literature to the best of our knowledge. " Unfortunately the authors ignore previous works on the subject where the transfer learning is applied to the NNN interactions. In fact, in [Computational Materials Science 198 (2021): 110702], and some references therein, a very similar procedure is utilized since the authors train the model outside of the critical region for the NN model and apply the trained model to classify the phases on the entire range of temperatures for the NNN case. The authors cannot ignore these previous works. A comparison with these papers must be added.
What is the meaning of "wild dataset"? To my knowledge, this is not a standard nomenclature. In fact, it would be quite confusing. I recommend avoiding the use of such terminology.
In Fig 2 the magnetization $\mu$ is normalized to 1, however, in figure 5 the authors show data corresponding to $\mu > 1$. How is this possible? What is exactly the data represented in Fig 5. I found this figure quite confusing.
Is the binary cross entropy maximized? What is the sense of this? Normally, I would expect a minimization of the BCE. This is a relevant point in the work that is poorly explained in the text.
The manuscript presents several typos throughout the entire document. A careful review of the grammar is necessary before submitting a new version

In summary, some of the questions presented in this manuscript may be of some interest for the machine learning methods applied to magnetic phase determination, however, the confusing presentation of the results, the absence of comparison with previous results in the literature concerning the transfer learning in the NNN Ising model and the many mistakes in the main text makes the answers to these questions unsatisfactory. For these reasons I cannot recommend the publication of the manuscript in the present form.

Requested changes

Section II should be rewritten to make it clearer and should contain a more detailed explanation of what they did.

Figure 5 does not correspond with the explanation the authors present throughout the text and with the other figures. The magnetisation is normalised to 1, but magnetisation values greater than 1 are shown. This figure should be corrected and a detailed explanation of what it contains should be added.

---

## Editorial Decision

awaiting_resubmission